# Maternal Psychological and Biological Factors Associated to Gestational Complications

**DOI:** 10.3390/jpm11030183

**Published:** 2021-03-05

**Authors:** David Ramiro-Cortijo, Maria de la Calle, Vanesa Benitez, Andrea Gila-Diaz, Bernardo Moreno-Jiménez, Silvia M. Arribas, Eva Garrosa

**Affiliations:** 1Department of Medicine, Beth Israel Deaconess Medical Center, Harvard Medical School, 330 Brookline Avenue, Boston, MA 02215, USA; dramiro@bidmc.harvard.edu; 2Department of Physiology, Faculty of Medicine, Universidad Autónoma de Madrid, C/Arzobispo Morcillo 2, 28029 Madrid, Spain; andrea.gila@uam.es; 3Obstetric and Gynecology Service, Hospital Universitario La Paz, Universidad Autónoma de Madrid, Paseo de la Castellana 261, 28046 Madrid, Spain; maria.delacalle@uam.es; 4Department of Agricultural and Food Chemistry, Faculty of Science, Universidad Autónoma de Madrid, C/Francisco Tomas y Valiente 7, 28049 Madrid, Spain; vanesa.benitez@uam.es; 5Pharmacology and Physiology PhD Program, Doctoral School, Universidad Autónoma de Madrid, C/Francisco Tomas y Valiente 2, 28049 Madrid, Spain; 6Department of Biological and Health Psychology, Faculty of Psychology, Universidad Autonoma de Madrid, C/Ivan Pavlov 6, 28049 Madrid, Spain; bernardo.moreno@uam.es

**Keywords:** anxiety, life satisfaction, life–work concerns, melatonin, cortisol, biopsychosocial model, obstetric complications

## Abstract

Early detection of gestational complications is a priority in obstetrics. In our social context, this is linked to maternity age. Most studies are focused on biological factors. However, pregnancy is also influenced by social and psychological factors, which have not been deeply explored. We aimed to identify biopsychosocial risk and protective factors associated with the development of maternal and fetal complications. We enrolled 182 healthy pregnant women, and plasma melatonin and cortisol levels were measured in the first trimester by chemiluminescent immunoassays. At different time points along gestation, women answered several questionnaires (positive and negative affect schedule, hospital anxiety and depression scale, pregnancy concerns scale, life orientation test, resilience scale, life satisfaction scale and life–work conflicts scale). They were followed up until delivery and categorized as normal pregnancy, maternal or fetal complications. Maternal complications were associated with low melatonin (OR = 0.99 [0.98; 1.00]; *p*-value = 0.08) and life satisfaction (OR = 0.64 [0.41; 0.93]; *p*-value = 0.03) and fetal complications were associated with high cortisol (OR = 1.06 [1.02; 1.13]; *p*-value = 0.04), anxiety (OR = 2.21 [1.10; 4.55]; *p*-value = 0.03) and life–work conflicts (OR = 1.92 [1.04; 3.75]; *p*-value = 0.05). We conclude that psychological factors influence pregnancy outcomes in association with melatonin and cortisol alterations. High maternal melatonin and life satisfaction levels could be potential protective factors against the development of maternal complications during pregnancy. Low anxiety and cortisol levels and reduced work–life conflicts could prevent fetal complications.

## 1. Introduction

Maternity is highly influenced by social factors, including education, economic or racial aspects. In industrialized societies, a key social determinant is the delay of maternity age [1], related to the gradual access of the women to higher education, employment and pregnancy control. The use of assisted reproduction techniques (ART) has made possible maternity beyond biological limits [2]. Pregnancy at an age over the optimum childbearing age has biological consequences, including higher rate of pregnancy complications and infertility. In addition, the use of ART increases the rates of multiple pregnancies [3], which are also a risk factor for complications, including preterm delivery (labor before 37 weeks of gestation) and fetal growth restriction (FGR) [4].

Almost 17% of pregnant women experience some type of pregnancy complication, which affects maternal and infant’s health [5]. Detecting women at risk as early as possible is a priority in obstetrics. Despite intensive research, there are still unknown causes that increase the risk of pregnancy complications. The majority of studies have focused on biological factors, some of which have been identified. However, pregnancy is influenced not only by biological determinants, but also by other factors. The biopsychosocial approach is ingrained in the “general systems theory”, which states that a system is characterized by the complex interactions of its components [6]. In the obstetric field, there is a psychological and social domain that should be incorporated into clinical practice to improve healthcare. Therefore, insight into pregnancy complications should be approached from a biopsychosocial point of view.

The psychological sphere likely exerts an important influence on pregnancy. Getting pregnant at an advanced age, the need of ART and a multiple pregnancy represent important stress factors which may negatively influence pregnancy outcome. There is evidence that individuals under high stress take poorer care of themselves and are more likely to engage in health-impairing behaviors, which may affect maternofetal health [7,8]. The importance of psychological factors on maternal health also is evidenced by the fact that antenatal optimism is the most protective factor against maternal postnatal adverse outcomes, e.g., depression or other mood disorders [9]. Maternal psychological factors may also influence long-term offspring health. For example, psychological stress in pregnancy, specifically in early stages, has been associated with higher risk of mental disorders in the offspring [10]. Life concerns, anxiety or problems in concealing work and personal life exert an influence on the biological environment of pregnant mothers and could be the link between psychological responses and negative influences on maternofetal health.

The interaction between psychological and biological factors and their impact on pregnancy outcomes have not been sufficiently explored [8]. Melatonin and cortisol are important hormones in gestation and can be potential biological factors implicated in this relationship. Melatonin is synthesized by the pineal gland with a peak at night [11], and during pregnancy it is also produced by the placenta [12]. Melatonin is a key hormone for pregnancy maintenance due to its pleiotropic roles [13], including its remarkable effects as an antioxidant [14]. We have evidence that, in the first trimester of pregnancy, a poor antioxidant status is linked with the development of pregnancy complications, and melatonin is an important contributor protecting against oxidative damage [3]. Our data indicate that maternal plasma melatonin is lower in women with preterm birth [15]. Daytime melatonin levels have been demonstrated to be influenced by anxiety and depression [16], perhaps due to the reduction in the number of sleep hours, which are directly proportional to melatonin circulating levels [17]. Therefore, it is possible that the dysregulation of this hormone under stressful conditions will influence pregnancy outcome.

Cortisol is another important hormone that may influence pregnancy outcome. Cortisol is synthesized by the hypothalamus–pituitary–adrenal axis and has cell catabolic and immune system suppression effects [18]. Cortisol secretion is closely linked to stress conditions, and it has been related to maternal mood, showing associations with psychological stress [19] and anxiety [20]. The maternal psychological stress and the associated increase in cortisol levels have been linked with adverse neonatal outcomes [21]. In this sense, we have evidence from twin pregnancies that the levels of maternal cortisol in early pregnancy are negatively associated with birth weight [15].

We hypothesize that social and psychological spheres exert an impact on obstetric outcome, influencing biological factors. We aimed to identify biopsychosocial risk and protective factors associated with the development of maternal and fetal complications in our social context, characterized by an advanced maternity age. We also explored the relationship between psychological variables and melatonin and cortisol, key hormones implicated in pregnancy and stress.

## 2. Materials and Methods

### 2.1. Cohort Selection

This is a retrospective, non-interventional and observational study from the Hospital Universitario La Paz (HULP, Madrid, Spain). Pregnant women were enrolled at 8 weeks of gestation at Obstetrics and Gynecology service. Women were first informed about the study proposal, and those who accepted anonymously and voluntarily to participate in the study signed a consent. The exclusion criterion was women who had previous diseases (hypertension, obesity, diabetes mellitus, inflammatory or immune deficiency diseases or record of previous pregnancy complications). The inclusion criterion was good comprehension of the Spanish language. Finally, 182 healthy pregnant women were enrolled. The study flow-chart is shown in Figure 1.

This study was performed in accordance with the Declaration of Helsinki regarding studies in human subjects and it was approved by HULP and Universidad Autónoma de Madrid Ethical Committees (PI-1490 and CEMU/2013-10, respectively).

### 2.2. Maternal and Neonatal Data Collection and Group Classification

Maternal and neonatal data were collected from the medical record. Maternal data collected were maternal age (years), civil status (single/married), educational level (undergraduate/university degree), employment situation (working/unemployment, of herself and, in the case of having a partner, also of the partner), family core-economic income (euros/month), smoking habits (yes/no) and alcohol intake during pregnancy (yes/no), gestational age (weeks of gestation), twin gestation (binary variable) and assisted reproduction techniques (ART, binary variable). In addition, obstetric adverse outcomes (diagnosed by the obstetrician staff based on hospital guidelines) were also recorded and included: *hyperemesis gravidarum* (defined as more than 3 episodes of vomiting/day, weight loss of 5% and ketones in the urine), pregnancy-induced hypertension (defined as blood pressure ≥ 140/110 mmHg without preeclampsia alterations before 20 weeks of gestation), gestational diabetes mellitus (defined as blood glucose levels > 140 mg/dL in the glucose tolerance test), preeclampsia (defined as pregnancy-induced hypertension and protein in the urine), pregnancy anemia (defined as hemoglobin < 11 g/dL in the first trimester or < 10.5 g/dL in the second or third trimester) and intrahepatic cholestasis of pregnancy (defined as pruritus and alterations in the blood/liver function tests including serum bile acids levels). Women who developed any of these complications during pregnancy were included in the “maternal complications” group.

Neonatal data recorded were infant sex and the following fetal adverse clinical outcome: any alterations in the physiologic systems reported by echography during pregnancy follow-up, FGR (defined as intrauterine growth below the third percentile or below the tenth percentile with hemodynamic alterations by Doppler echography; binary variable) and preterm labor (defined as gestational age low than 37 weeks; binary variable). Pregnancies with any of these fetal outcomes were included in the “fetal complications” group.

### 2.3. Maternal Plasma Variables in the First Trimester

At 9–11 weeks of gestation, blood samples were extracted from 8:00 to 9:00 am, by venipuncture in Vacutainer^®^ lithium heparin gel tubes for plasma separation (Becton Dickinson comp., Madrid; Spain), following the protocols established by the medical staff. Plasma was obtained by centrifugation (2100× *g*, 15 min at 4 °C), within a maximum of 2 h after extraction and stored at −80 °C to assay melatonin and cortisol.

To determine melatonin levels, plasma was evaporated to dryness with an evaporator centrifuge (Speed Vac SC 200; Savant; Hyannis, MA, USA). The residues were dissolved in distilled water and melatonin levels were determined by a competitive enzyme immunoassay kit (Melatonin ELISA; IBL International, Hamburg; Germany) according to the manufacturer’s instructions. The kit is characterized by an analytical sensitivity of 1.6 pg/mL and high analytical specificity (low cross-reactivity). Melatonin was expressed as pg/mL. This method was extensively described in Aguilera et al. [22].

To assess plasma cortisol, a competitive immunoassay using direct chemiluminescent technology was used, being analyzed with an Advia Centaur instrument (Siemens Healthineers, Madrid; Spain). Cortisol was expressed as µg/dL.

### 2.4. Maternal Psychological Variables during Pregnancy

Women answered psychological applications in each trimester of pregnancy at Weeks 9–11 (first trimester), 24–25 (second trimester) and 36 (third trimester), during a programmed visit to the hospital. Missing data referred to women with miscarriage, preterm delivery or follow-up loss, coinciding with the visit of pregnant women to the hospital. Psychological applications are described below:

*Positive and Negative Affect Schedule* [23], Spanish version [24]. This standardized application scores the positive and negative affectivity using 20 items with a Likert answer of 5 points ranging from “nothing” (1) to “extremely” (5). Some of the positive items were “active” or “strong” and some of the negative items were “restless” or “distressed”.

*Hospital Anxiety and Depression Scale* [25], Spanish version [26]. This standardized application scores the anxiety using a questionnaire with seven items with a Likert answer of 4 points ranging from “no anxiety” (0) to “high anxiety” (3).

*Pregnancy concerns scale*. It is an ad-hoc scale, which was elaborated according to the recommended phases for the creation of psychological scales [27] and was previously used by our research group [28]. This scale evaluates personal concerns about gestation worries. It is a scale with 10 items with a Likert answer of 4 points ranging from “none” (0) to “many” (3). Some items were “the health of my newborn” or “the effect of the drugs on the fetus”.

*Life Orientation Test* [29], Spanish version [30]. This application scores personal optimism. We used a short scale of five items with a Likert answer of 5 points ranging from “completely agree” (1, low optimism) to “completely disagree” (5, high optimism). Some examples of the items are “I am always optimistic about my future” and “I usually think the things are not going right”.

*Resilience scale* [31], Spanish version [32]. This application scores the ability to cope with daily difficulties/problems. We used a short scale of six items with a Likert answer of 7 points ranging from “completely disagree” (1, low resiliency) to “completely agree” (7, high resiliency). Some items in the scale were “In any situation, I can be efficient” and “When I believe in myself, I can go through difficult times”.

*Life Satisfaction scale* [33], Spanish pregnancy version [34]. This application scores subjective satisfaction with your own life. The scale has five items with a Likert answer of 5 points ranging from “completely disagree” (1, low life satisfaction) to “completely agree” (7, high life satisfaction). Some items in the scale were “In general, my real life is close to my ideal life” or “If I were born again, I don’t think I’d change anything in my life”.

*Life–work conflicts scale*. It is an ad-hoc scale, which was elaborated according to the phases for the construction of psychological scales [27] and was previously used by our research group [28]. This application scores the association between work and family life difficulties. It is a scale with six items with a Likert answer of 4 points ranging from “never” (0, minimum conflicts) to “always” (3, maximum conflicts). Some items were “I am irritable at home because my work is exhausting” and “I have to cancel plans with my partner/family/friends due to work commitments”.

The points of data collection and reliability of psychological applications are summarized in Table 1.

### 2.5. Statistical Analysis

Statistical analysis was performed using R software (version 3.6.0, R Core Team, 2018, Vienna; Austria) within R Studio interface, using MASS, pscl, oddsratio, ggplot2, ggpubr and cowplot packages. Continuous variables were expressed as median and interquartile range (IQR) and qualitative variables as relative frequency. The univariate analysis for the maternal characteristics, plasma and psychological variables were compared using the Mann–Whitney test. The correlation between hormonal levels and psychological factors was analyzed by the rho-Spearmen test. The association between quantitative variables was assessed by the Chi-squared test. Those variables that showed a *p*-value < 0.10 in the univariate analysis were considered as modulatory variables in the regression models. This criterion was proved as the most parsimonious analysis in the models [35].

Regression models were created through stepwise procedures to estimate the major contribution of maternal/neonatal characteristics, maternal plasma and psychological variables to maternal and fetal complications. We conducted a stepwise analysis of potential confounders in five sequential categories: (1) maternal characteristics: maternal age, gestational age, twin gestation, ART, infant sex, FGR and preterm labor; (2) maternal plasma variables at first trimester: melatonin and cortisol; (3) psychological variables at first trimester; (4) psychological variables at second trimester; and (5) psychological variables at third trimester. The regression models show adjusted odds ratios (OR) with 95% of confidence interval (CI) and determination coefficients (R^2^). In the regression models, the collinearity variables were discharged. The *p*-value < 0.05 was considered significant. In addition, *p*-value < 0.10 was reported as a quasi-significant trend.

## 3. Results

The prevalence of maternal complications in the cohort was 46.2% (84/182) and the prevalence of fetal complications was 25.0% (45/182). There was no statistical association between maternal and fetal complications (*χ*^2^ = 0.759; *p*-value = 0.38).

### 3.1. Maternal Characteristics and Development of Obstetric Complications

Overall, 67.5% (123/182) of the women in the study had graduate degree, being the rest undergraduate degree. There were no women with less than an undergraduate education. Overall, 67.0% (122/182) had a partner; and the partner was employed for 95.1% (116/122). Overall, 86.2% (157/182) of the women were employed during the study period. Regarding the economic level of the family core, 74.1% (135/182) of the cohort earned more than 2000 euros/month.

We did not find statistical differences in any of the socioeconomic parameters analyzed (civil status, educational level, economic status and alcohol or tobacco consumption) between women with and without maternal complications. No differences were found in relation to either infant sex or maternal age. The group of women with maternal complications showed a significantly lower gestational age and higher use of ART, twin pregnancies and preterm labor, compared to women without maternal complications (Table 2).

Regarding differences between women who developed or not fetal complications, no statistical differences were found in socioeconomic parameters or infant sex. However, women with fetal complications had significantly higher age, use of ART, twin pregnancies, preterm labor and FGR, compared to women without fetal complications (Table 2).

In our cohort, ART and twin pregnancy were significantly associated (*χ*^2^ = 79.512; *p*-value = 0.001), as well as preterm labor and FGR (*χ*^2^ = 4.391; *p*-value = 0.036). Based on these collinearities, the variables twin pregnancy and preterm labor were included in the logistic regression models.

### 3.2. Maternal Plasma Variables in the First Trimester

In pregnancies without any obstetric adverse outcome, plasma melatonin levels were 16.0 (27.3) pg/mL. Melatonin levels tended to be lower in women who developed a maternal complication compared to those who did not develop a maternal complication, it but did not reach statistical significance (*p*-value = 0.09; Figure 2A). Melatonin levels were not significantly different between women with and without fetal complications (Figure 2B).

In pregnancies without any obstetric adverse outcomes, plasma cortisol levels were 19.2 (8.7) μg/dL. Cortisol levels did not show statistical differences between women with and without maternal complications (Figure 2A). Cortisol tended to be higher in the group who developed fetal complications but did not reach statistical significance (*p*-value = 0.07) (Figure 2B).

### 3.3. Psychological Variables during Pregnancy

Psychological variables were reported according to the gestational trimester. Table 3 shows the scores of the study groups and mothers without any obstetric adverse outcome (maternal or fetal complications).

Regarding maternal complications, in the first trimester, we did not detect statistical differences between groups in any of the parameters evaluated. In the second trimester, women who developed maternal complications showed significantly lower scores in life satisfaction compared to those without maternal complications. In the third trimester, the group of maternal complications scored statistically lower in both life satisfaction and resilience than women without maternal complications (Table 3).

Regarding the group of fetal complications, in the first trimester, women with fetal complications showed a significantly higher score in anxiety compared to women without fetal complications. In the second and third trimesters, no statistical differences in the psychological variables were detected between mothers with and without fetal complications (Table 3).

In the cohort (n = 182), we found in the first trimester a significant and positive correlation between resilience and melatonin levels (rho = 0.18; *p*-value = 0.021). Although without significant differences, in the second trimester, we found a negative correlation among negative affect (rho = −0.15; *p*-value = 0.09), pregnancy concerns (rho = −0.15; *p*-value = 0.07) and melatonin levels. Furthermore, in pregnant women with maternal and fetal complications, we found in the second trimester a significant and positive correlation between positive affect and melatonin levels (rho = 0.55; *p*-value = 0.033). However, we did not find a statistical correlation between cortisol levels and any of the psychological scores.

Since a high proportion of women in the study had pregnancies derived from ART (*n* = 66), we also analyzed differences in psychological variables according to use of ART. In the first trimester, ART had significantly higher scores of negative affect (ART = 2.1 (1.0); non-ART = 1.7 (0.8), *p*-value = 0.026) and anxiety (ART = 1.1 (0.6); non-ART = 0.7 (0.4), *p*-value = 0.006). In the third trimester, ART also had significantly higher scores of negative affect (ART = 2.3 (1.1); non-ART = 2.0 (1.0), *p*-value = 0.010) and anxiety (ART = 1.3 (0.8), non-ART = 1.0 (0.7), *p*-value = 0.007). Furthermore, ART had significantly lower scores of positive affect (ART = 3.4 (0.7); non-ART = 3.5 (0.8), *p*-value = 0.033) and life satisfaction (ART = 5.4 (1.6); non-ART = 5.8 (1.0), *p*-value = 0.017).

### 3.4. Logistic Regression Models Associated with Maternal and Fetal Complications

The models showed that maternal complications were associated with preterm labor, indicating that maternal complications are a risk factor for prematurity. Data also evidence that melatonin levels in the first trimester and the psychological variable “life satisfaction” in the third trimester were negatively associated with maternal complications and, therefore, could be considered protective factors (Figure 3A).

In the models of fetal complications, preterm labor was a risk factor associated with fetal complications. Other risk factors for fetal complications were maternal cortisol levels and anxiety in the first trimester, and life–work concerns in the second trimester (Figure 3B).

## 4. Discussion

The objective of this study was to identify biopsychosocial risk and protective factors associated with the development of adverse obstetric outcomes. Our data show that maternal psychological features exert an influence on pregnancy outcomes; in particular, high scores in life satisfaction could be a protective factor to prevent maternal complications, while anxiety and life–work conflicts may be risk factors of fetal complications. The present study also points out the relationship between maternal melatonin and cortisol levels in early pregnancy and obstetric outcomes. We explored the association between maternal psychological parameters and these plasma hormones. Although we did not find a significant correlation with cortisol, we found some associations between psychological variables and melatonin. Melatonin levels were positively associated with resilience in the first trimester and with positive affect in the second trimester, while a negative association was found with pregnancy concerns. In summary, our data reveal the relationship between social, psychological and biological spheres, which can exert an influence on pregnancy outcomes. Our data point out the need to study pregnancy from a more global biopsychosocial approach.

Pregnancy complications are increasing in our society due to several factors and represent an important health problem. Thus far, they have been studied mainly from a clinical point of view. The incorporation of the psychological and social domain into clinical practice may help to understand their origin and to improve healthcare. The relevance of considering multiple spheres to predict the clinical outcomes has been demonstrated in perinatology [36]. In this work, we considered this global approach, analyzing the social, psychological and biological spheres. These different aspects are discussed below.

*Social factors.* Regarding the social sphere, poverty and racial disparities have usually been explored as key determinants of health. In the studied population, these factors do not seem to play an important influence on the obstetric and neonatal outcome, since the population studied is of middle-high socioeconomic level. A relevant characteristic in our population was the age, which was above the optimum for maternity. This fact reflects a key aspect in high-income countries: the delay in the maternity age. In our social context, there has been a gradual access of the women to higher education and employment, as well as in pregnancy control [37]. These sociocultural determinants may be some of the factors implicated in the continuing rise in the age of childbearing observed from the second half of the twentieth century [38]. High-income countries enacted comprehensive maternity legislation providing women with rights, such as a period of employment protection for childbirth [39]. Legislation can help solving the problem of work–family conflicts; however, delayed maternity age and its impact on women’s health remains a problem. In fact, in our population, we found that maternity age was associated with adverse fetal outcomes.

The association between advanced maternity age and adverse outcome may have biological grounds. On the one hand, it is linked to infertility. ART has made possible maternity beyond biological limits. However, the main consequence is an increased rate of twin pregnancies, which are a risk factor for obstetric complications, particularly preterm labor [38,40]. This was confirmed in the present study, evidencing that twin pregnancies were nearly seven times more likely to develop a complication compared to singletons. It is important to note that obstetric adverse outcomes are not a direct effect of ART, but rather of the fact that the use of ART is associated with twin pregnancies. In fact, it has been found that the rate of obstetric complications in twin pregnancies derived from ART is not higher than in spontaneous twin pregnancies, even at 45 years old [41]. In addition, an important aspect to explore would be the relationship of particular ART techniques and obstetrical outcomes, which we did not collect it.

In addition to the biological influence on pregnancy outcome, psychological factors were considered. Getting pregnant at an advanced age, the process of repetitive ART cycles and a multiple pregnancy represent important stress factors for the mother, which may exert negative influences. Therefore, the psychological sphere should be considered.

*Psychological factors*. Psychological stress during pregnancy affects the maternal–fetal binomial health and is known to have effects on pregnancy hypertension, fetal programming and gestational timing [42]. According to the literature, pregnancy is a period of significant life change for a woman and her partner, and it can be perceived as a stressful situation. Research has found that negative life events were associated with an increased risk of fetal complications and emotional distress in the mother [43]. Our data support that pregnant women with high negative affect scores are associated with maternal and fetal complications. In the same line, other studies have found that psychological optimism, pessimism or anxiety are associated with birth outcomes [44]. Our data show that maternal anxiety in the first trimester is a risk factor for later development of fetal complications.

Pregnancy represents a challenge, and, for many women, it may be a stressful period, particularly if there are insufficient psychosocial resources or additional stressors. One of them is the use of ART, which implies a high level of uncertainty. We therefore explored the possible influence on psychological factors and pregnancy outcomes. As expected, we found higher scores of negative affect and anxiety and lower positive affect and life satisfaction in pregnancies derived from ART. If the response to a stressor is inadequate, it may lead to a negative influence on health. Our data show that ART-derive pregnancies were at higher risk of maternal and fetal complications. This has been proposed to be directly linked to advanced maternity age and the association with twin pregnancies; however, our data reveal high levels of anxiety, which may also contribute to the worse outcome and could be taken into consideration.

According to the Commission on the Social Determinants of Health from the World Health Organization [45], a stressful workplace is considered one of the most important psychological stressors, while stable family core and social behaviors are key positive aspects for mental health [46]. Our data support that pregnant women with maternal and fetal complications scored low in life satisfaction and high in life–work concerns. Today, health professionals should be aware of the influence of psychological processes and social behavior disadvantage on disease; in fact, sociodemographic disadvantages are also postulated as independent risk factors for adverse pregnancy outcomes [47].

It is important to note that the psychological scores in the second and third trimesters may be biased because pregnancy problems have already been diagnosed and maternal behaviors could be affected.

*Biological factors*. Pregnancy is a period of biological changes, which interacts with the psychological sphere. A stressful job, the problems associated with the pressure of getting pregnant at an advanced age, the use of ART and twin pregnancies are important psychological factors which may influence the biological domain [43]. Even though the biological milieu has been thoroughly studied in relation to obstetric complications, the relationship between psychological and biological spheres has not been fully addressed before. In our study, we assessed two key hormones, melatonin and cortisol, since they are relevant in pregnancy and have been previously shown to be affected by psychological conditions.

Melatonin has been associated with the psychological alterations, and it has been proposed that low levels of this hormone may underlie the pathophysiology of depression and other mood disorders, which are also relatively frequent the context of pregnancy. In addition, this hormone is relevant in the maintenance of a normal pregnancy. Therefore, we hypothesized that melatonin could be a keystone between psychological processes and pregnancy disorders. Our data show a negative correlation between melatonin levels and pregnancy concerns. In addition, we demonstrated that women with maternal complications tended to have lower melatonin levels. This association may be related to poorer antioxidant defenses, since oxidative stress is a key mechanism in pregnancy complications [48] and the role of melatonin as protective hormone in pregnancy is related to its powerful antioxidant actions [14]. In fact, previous data of our group demonstrate that low plasma melatonin was associated with low levels of antioxidants and development of maternal complications [3] and with preterm delivery in twin gestation [15]. We suggest that low melatonin can contribute to a poor antioxidant balance in early pregnancy and participate in pregnancy complications later on. Melatonin is a hormone with a circadian rhythm and its highest levels occur during sleep. Therefore, the reasons for low melatonin levels can be directly related to sleep deficiency. Sleep disturbances can be linked to psychological factors. For example, it has been shown that life–work conflicts and pregnancy concerns are key factors influencing sleep [48]. According to our data, pregnancy concerns tended to be increased in women with maternal complications. Therefore, it is possible that the association between low melatonin and poor pregnancy outcome could be partially related to a reduction of the number of sleep hours. Additionally, the effects of job-related stressors on work–family conflict are most often viewed from the perspective of conservation of resources theory [49]. According to the theory, individuals have a finite amount of time, attention and energy, and, therefore, higher time commitment or demand from one role puts pressure on other roles [50,51]. We analyzed the possible relationship between melatonin and resilience in the first trimester and found a positive association. We propose that melatonin can be a hormone affected by psychological variables, being associated with positive and negative mood in pregnancy. This hypothesis needs to be studied further since the present work had some limitations, such as assessing melatonin at daytime and at a single and point. It would be desirable to analyze also night levels along pregnancy, together with evaluation of sleep quality and psychological variables.

Cortisol is a hormone with multiple physiological roles and well known to be secreted in stressful situations. In pregnancy, maternal cortisol passage to the fetus is limited by the placental barrier 11β-dehydrogenase enzyme (11β-HSD-2), which transforms cortisol into cortisone, an inactive glucocorticoid [52,53]. It has been demonstrated that 11β-HSD-2 is inhibited as a consequence of maternal psychological stress, affecting fetal growth [54,55]. There is also evidence that pregnant women who had anxiety showed a reduction of uterine blood flow [56] which is related to FGR. These studies are in accordance with our data showing high anxiety scores in women who developed fetal complications together with a trend towards high levels of cortisol in the first trimester. Our data add evidence to the link between maternal stress during early pregnancy and adverse maternal-fetal outcome, through alterations in the hormonal milieu. However, although we analyzed the possible association between cortisol levels and the psychological variables, such as anxiety, we did not find any potential relationship. Our study had some limitations regarding cortisol measurements, which could account for this. Firstly, the assessment at a single time point in the morning; evaluation of the hormone at different time points would have been more informative, allowing analyzing other parameters such as the diurnal cortisol slope and cumulative cortisol output across the day. It is also possible that venipuncture could have caused additional stress and cortisol release; this effect has been mainly observed in children related to fear of pain and is likely minimized in our population of adults with a programmed intervention. Cortisol measurements in saliva could avoid this possible problem.

### Strengths and Limitations

The main strength of this study was the longitudinal approach in the psychological variables. This condition adds power to the design compared to cross-sectional studies, by virtue of observing the temporal order of events. The second strength of the study was the exploration of multiple spheres in the pregnancy context. The new methodological approaches to research in human biomedicine must consider the biological, psychological and social points of view. Thirdly, the logistic regression models with the stepwise method allow estimating the contribution of biological, psychological and social areas sequentially.

Regarding limitations, the first is the assessment of the biochemical variables only at a one-time point since cortisol and melatonin exhibit circadian rhythms. This was due to ethical limitations, which restricted the blood collection only in the morning and in the first trimester, coincident with a routine analysis. Although blood samples were extracted in the same conditions and the levels were determinate under the same protocols for all women, this aspect needs to be taken into account in future studies. The second limitation is the sample size, which is smaller in the third trimester than in the first one, related to prematurity, stillbirths or loss to follow-up, resulting in fewer women completing the psychological applications at the end of pregnancy. It would also be desirable to have a more heterogeneous population reflecting other social environments, since the present sample is homogeneous in terms of age, education and income. Thirdly, the analysis of the biological markers at different time points along gestation, and the inclusion of other molecules such as inflammatory mediators, oxidative stress markers or placental and vascular growth factors, known to be related to pregnancy disorders would help to complete the relationship between biological and psychological variables and their role in pregnancy outcomes.

In addition, it would be necessary to carry out new studies to check whether the results are maintained in other cohorts with another methodologies (i.e., Monte Carlo simulation analysis). However, the key element is to start from a theoretical model that justifies the analyses performed, not only focus on the method [57]. Gaining knowledge on factors from these three spheres would help building up the biopsychosocial model of pregnancy, which would aid maternal counseling to improve pregnancy outcomes.

## 5. Conclusions

In this work, using logistic regression models, we detected a relationship between some maternal psychological and biological factors (melatonin and cortisol) and the development of complications.

To help preventing maternal complications, high melatonin levels at the beginning of gestation and life satisfaction in mid-pregnancy could be protective factors. Our results also suggest that low anxiety and cortisol levels at the beginning of pregnancy and reducing problems of work–life conciliation could help prevent fetal complications. Considering these data, it may be interesting to promote health public guidelines to support resources, which will improve life satisfaction and life–work conciliation during pregnancy. In addition, to know the social-reality of each pregnant women, intervening from primary care to specialized clinical psychology would help humanize pregnancy. These policies contribute to the decline in the rate of adverse obstetric outcomes with a direct impact on the healthcare cost, particularly in societies with advanced maternity age.

## Figures and Tables

**Figure 1 jpm-11-00183-f001:**
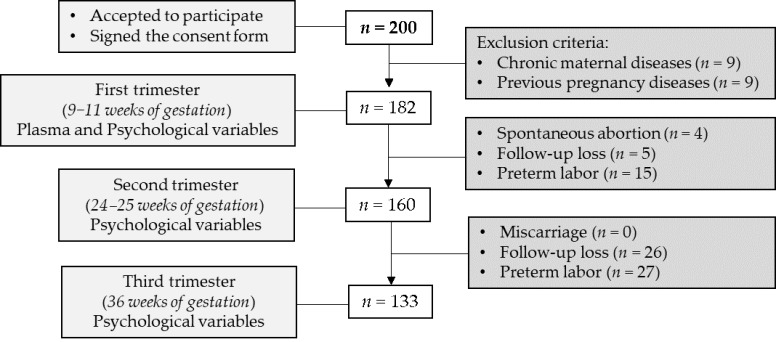
Flow-chart of study participants. Exclusion criteria: chronic maternal disease (i.e., hypertension, obesity, diabetes mellitus, inflammatory or immune deficiency diseases previous pregnancy) and diagnoses of obstetrical complications before. Sample size (*n*) is shown between brackets.

**Figure 2 jpm-11-00183-f002:**
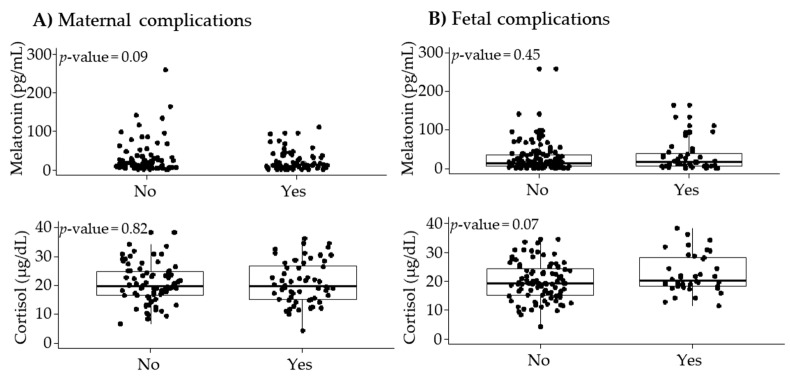
Maternal plasma melatonin and cortisol levels according to maternal (**A**) and fetal (**B**) complications. Data show median and interquartile range (IQR). The *p*-value was obtained by Mann–Whitney U test.

**Figure 3 jpm-11-00183-f003:**
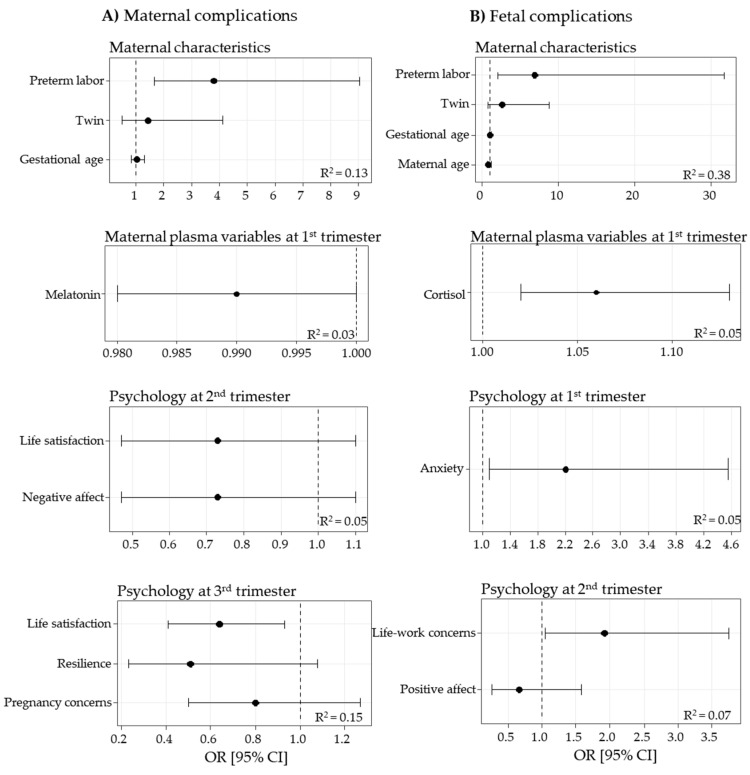
Logistic regression models, obtained by stepwise procedures, for maternal (**A**) and fetal (**B**) complications. Data show odds ratio (OR) [95% confidence interval (CI)] and determination coefficients (R^2^).

**Table 1 jpm-11-00183-t001:** Timing for data collection and reliability of the psychological application.

	Cronbach’s α	1T	2T	3T
Maternal and neonatal data	-	X	X	X
Maternal plasma variables	-	X		
**Maternal psychological variables**				
Negative affect	0.80	X	X	X
Positive affect	0.73	X	X	X
Anxiety	0.84	X	X	X
Pregnancy concerns	0.82	X	X	X
Optimism	0.66	X		X
Resilience	0.66	X		X
Life satisfaction	0.87		X	X
Life–work conflicts	0.88		X	

First trimester (1T), second trimester (2T) and third trimester (3T) of pregnancy. Cronbach’s α was used to assess the reliability of psychological applications in this study.

**Table 2 jpm-11-00183-t002:** Maternal characteristics according to maternal and fetal complications.

	Maternal Complications			Fetal Complications		
	No (*n* = 98)	Yes (*n* = 84)	*p*-Value	No (*n* = 125)	Yes (*n* = 45)	*p*-Value
Maternal age (years)	35.0 (6.0)	35.0 (7.0)	0.51	34.0 (6.0)	35.0 (5.0)	0.039
Civil status						
Single	26.6% (21)	26.7% (24)	0.40	28.3% (33)	27.5% (11)	0.87
Married	73.4% (58)	71.1% (64)		71.8% (84)	72.5% (29)	
Educational level						
Undergraduate	21.0% (17)	34.5% (32)	0.49	27.3% (33)	29.2% (12)	0.77
Graduate	77.7% (63)	64.5% (60)		71.1% (86)	70.7% (29)	
Employment situation						
Working	93.8% (76)	87.1% (81)	0.16	91.7% (111)	85.4% (35)	0.25
Unemployment	6.2% (5)	9.7% (9)		5.7% (7)	14.6% (6)	
Smoking habits	14.8% (12)	17.2% (16)	0.61	14.0% (17)	19.5% (8)	0.35
Alcohol intake	42.0% (34)	39.1% (36)	0.70	38.3% (46)	48.8% (20)	0.24
Gestational age (weeks)	38.0 (3.0)	37.5 (2.0)	0.006	38.0 (2.4)	35.8 (2.5)	0.001
Twin pregnancies	38.1% (37)	67.5% (56)	0.001	41.9% (52)	80.0% (36)	0.001
ART	25.5% (25)	49.4% (41)	0.001	28.0% (35)	60.0% (27)	0.001
Male ^1^	52.3% (46)	60.8% (48)	0.27	57.6% (68)	51.3% (20)	0.49
Male ^2^	41.9% (13)	52.9% (27)	0.33	54.2% (26)	42.9% (12)	0.34
Preterm labor	17% (16)	32.9% (26)	0.015	0% (0)	52.4% (22)	0.001
Fetal growth restriction	4.3% (4)	12.2% (9)	0.060	0% (0)	33.3% (15)	0.001

Data show median and interquartile range (IQR) in quantitative variables and relative frequency (*n*) in qualitative variables. ^1^ Sex in single and first newborn in twin pregnancies. ^2^ Sex of the second newborn in twin pregnancies. Assisted reproduction techniques (ART). The *p*-value was obtained by Mann–Whitney U and Chi-squared tests, for quantitative or qualitative variables, respectively.

**Table 3 jpm-11-00183-t003:** Psychological variables according to maternal (*n* = 84) and fetal (*n* = 45) complications.

	First Trimester							Second Trimester							Third Trimester						
	Ref.	Maternal Complic.			Fetal Complic.			Ref.	Maternal Complic.			Fetal Complic.			Ref.	Maternal Complic.			Fetal Complic.		
		No	Yes	*p*-Value	No	Yes	*p*-Value		No	Yes	*p*-Value	No	Yes	*p*-Value		No	Yes	*p*-Value	No	Yes	*p*-Value
Negative affect	1.7 (0.7)	1.8 (0.8)	2.0 (1.0)	0.16	1.8 (0.8)	2.1 (0.7)	0.13	1.7 (0.8)	1.8 (0.8)	2.0 (0.7)	0.054	1.8 (0.9)	2.0 (0.8)	0.12	1.9 (1.1)	1.9 (1.1)	2.1 (1.0)	0.15	2.0 (0.9)	2.1 (1.1)	0.49
Positive affect	3.4 (0.7)	3.4 (0.7)	3.3 (0.8)	0.76	3.3 (0.8)	3.3 (0.9)	0.63	3.5 (0.8)	3.4 (0.8)	3.4 (0.8)	0.80	3.4 (0.7)	3.2 (0.7)	0.056	3.6 (0.7)	3.5 (0.7)	3.5 (0.8)	0.36	3.5 (0.8)	3.5 (0.7)	0.81
Anxiety	0.9 (0.6)	0.9 (0.7)	0.9 (0.7)	0.90	0.9 (0.6)	1.1 (0.7)	**0.041**	1.0 (0.9)	1.0 (0.9)	1.0 (0.9)	0.62	1.0 (0.7)	1.1 (1.1)	0.18	0.9 (0.7)	1.0 (0.7)	1.1 (1.0)	0.14	1.0 (0.9)	1.1 (1.0)	0.36
Pregnancy concerns	1.6 (0.1)	1.6 (0.7)	1.7 (0.7)	0.96	1.7 (0.8)	1.6 (0.7)	0.52	1.8 (0.1)	1.8 (0.9)	1.8 (0.8)	0.68	1.7 (0.8)	1.9 (0.8)	0.78	1.9 (0.1)	1.9 (0.7)	1.8 (0.8)	0.063	1.9 (0.8)	1.8 (0.8)	0.83
Optimism	3.4 (0.9)	3.4 (1.4)	3.2 (1.2)	0.28	3.2 (1.2)	3.6 (1.4)	0.28	-	-	-	-	-	-	-	3.4 (0.9)	3.4 (0.9)	3.2 (1.0)	0.49	3.2 (0.8)	3.5 (1.0)	0.17
Resilience	6.0 (0.1)	6.2 (0.7)	6.0 (1.0)	0.20	6.0 (0.8)	6.2 (0.7)	0.35	-	-	-	-	-	-	-	3.0 (0.1)	6.0 (0.9)	5.3 (1.8)	**0.001**	5.7 (1.2)	5.7 (1.5)	0.47
Life satisfaction	-	-	-	-	-	-	-	5.8 (0.9)	5.8 (0.9)	5.5 (1.0)	**0.015**	5.8 (1.1)	5.6 (1.2)	0.59	5.8 (0.8)	5.8 (1.0)	5.4 (1.2)	**0.010**	5.8 (1.0)	5.6 (1.8)	0.61
Life–work conflicts	-	-	-	-	-	-	-	0.8 (0.1)	0.8 (1.0)	1.0 (1.3)	0.70	0.8 (1.2)	1.2 (1.2)	0.061	-	-	-	-	-	-	-

Data show median and interquartile range (IQR). Reference data (Ref.) show the psychological score in our cohort in pregnancies without any obstetric adverse outcomes (*n* = 72). Complications (complic.). The *p*-value was obtained by Mann–Whitney U test.

## Data Availability

The raw datasets used for this study contained health personal information and the Ethical Committee requirements has forbidden the data transfer. However, a particular report could be sent to the corresponding authors by email.

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
