# Peer review of "Maternal Psychological and Biological Factors Associated to Gestational Complications"

_jpm, 2021, doi:10.3390/jpm11030183_

Round 1

Reviewer 1 Report

Thank you for your contribution to this important topic. This reviewer has some comments/suggestions.

-The very first sentence in the introduction is confusing and needs to be revised.

-Although important, the introduction, especially the first 3 paragraphs, is not well organized and do not flow well together. It was difficult to identify the overall purpose of the study. It was not clear what the study was going to be about just reading the introduction.

-Please remove "Besides" as a introductory word in all sentences.

-Figure 1: the footnote does not seem to be complete. Are those exclusion criteria listed?

-Pregnancy Concerns Scale and Life-Work conflicts: are these author created? They do not have a reference. If they are new scales, it is best to discuss validity and reliability of how these scales were developed.

-Was a power analysis done? On what aim?

-Table 3 is very difficult to read and needs to be revised; the horizontal lines are off

-Line 364: this conclusion does not quite match the findings in Table 2 and 3. This can discredit what the authors are trying to convey if the data does not match the conclusions/summary.

-There is so much data to discuss and the discussion section was poorly written and needs more organization and clarity.

-line 384, if this is the theoretical foundation for the study, it is recommended to include in the introduction or methods as a guide to your study.

Author Response

Response: thank you for taking the time to review our article. Please, see below our responses to improve the article.

  • The very first sentence in the introduction is confusing and needs to be revised.

Response: the first sentences have been modified.

  • Although important, the introduction, especially the first 3 paragraphs, is not well organized and does not flow well together. It was difficult to identify the overall purpose of the study. It was not clear what the study was going to be about just reading the introduction.

Response: We have modified these paragraphs in the introduction. We hope to provide a better idea of the study.

  • Please remove "Besides" as a introductory word in all sentences.

Response: according to the reviewer’s comments, we have eliminated the “besides”.

  • Figure 1: the footnote does not seem to be complete. Are those exclusion criteria listed?

Response: the footnote in the flow-chart was modified.

  • Pregnancy Concerns Scale and Life-Work conflicts: are these author created? They do not have a reference. If they are new scales, it is best to discuss validity and reliability of how these scales were developed.

Response: these scales were created by the research group (mentioned in the text "Ad-hoc") and were elaborated according to the recommendations procedure for the creation of a new psychological scale. We have modified the text to clarify this aspect and added reference 27.

  • Was a power analysis done? On what aim?

Response: This was a non-interventional, observational study. Since all women were healthy at baseline and we did not randomize to treatment arm, it was not necessary to determine the type II error a priori but to establish rigorous criteria for the type I error (p-value).

  • Table 3 is very difficult to read and needs to be revised; the horizontal lines are off.

Response: We agree with this comment, and although it is a journal guideline format, we have tried to include more lines and shading in the reference column to improve reading.

  • Line 364: this conclusion does not quite match the findings in Table 2 and 3. This can discredit what the authors are trying to convey if the data does not match the conclusions/summary.

Response: Tables 2 and 3 represent only the univariate analysis with the comparison in social, clinical, and psychological variables between groups. However, the strength of the study and the conclusions derived from the adjusted logistic regression models (represented in figure 3), which evidenced associations between some of the psychological and biological variables analyzed and the development of maternal or fetal complications. 

  • There is so much data to discuss and the discussion section was poorly written and needs more organization and clarity.

Response: we agree with the reviewer’s comment and we have re-oriented deeply the discussion.

  • line 384, if this is the theoretical foundation for the study, it is recommended to include in the introduction or methods as a guide to your study.

Response: we have added some information regarding the theory in the introduction, for clarification (lines 50-55).

Reviewer 2 Report

Overall this paper is very interesting and novel, but I have a few concerns/questions. The authors include a wide range of relevant outcomes and assessments. The background/rationale is well presented and eloquently integrates many key concepts from different domains. This article would be of interest to researchers and clinicians alike interested in maternal/perinatal health, neuroendocrinology, and psychology.

Some minor proofreading is needed (e.g. frequent use of the word "besides" to start sentences) some awkward sentences, especially in the beginning of the methods. 

I am confused about the population being studied. Why were the women seen in a high risk setting, but also previous illnesses excluded? What criteria is there for them to be considered high risk but also not have a previous illness? This affects generalizability to other samples.

Line 130: why is "(undergraduate/university degree)" the only option for education level?

Were smoking and alcohol considered separately or was saying yes to either one considered a "yes" for the entire variable ?

I am skeptical of the use of stepwise regressions: see e.g. Smith (2018) https://journalofbigdata.springeropen.com/articles/10.1186/s40537-018-0143-6 I don't think the paper can be accepted unless this is changed. Select your covariates based on previous literature/mechanistic basis, etc.

Was there any correction for multiple tests done, especially with a relatively small sample size? e.g. false discovery rate? If not, why?

Overall, I still think the paper needs some revisions. The content is very interesting and can inform future perinatal/maternal health research. I applaud the authors for their thorough consideration of biopsychosocial factors.

Author Response

Response: thank you for taking the time to review our article. Following your recommendations, we have made some changes to improve the article.

Some minor proofreading is needed (e.g. frequent use of the word "besides" to start sentences) some awkward sentences, especially in the beginning of the methods.

Response: according to the reviewer’s comments, we have reduced the “besides” in the text.

I am confused about the population being studied. Why were the women seen in a high-risk setting, but also previous illnesses excluded? What criteria is there for them to be considered high risk but also not have a previous illness? This affects generalizability to other samples.

Response: the reviewer's comment is interesting and deserves an explanation. Twin pregnancies are considered high-risk pregnancies, but women are healthy otherwise (at least at the beginning of pregnancy). These pregnancies (and sometimes women with advanced maternal age or ART-derived pregnancies) are attended from the primary health centers to the so-called high-risk units at the reference hospital for a better follow-up. The unit attends also non-risk pregnancies.  This is the studied cohort.  Our criterion was that in week 10 of pregnancy (when we took a blood sample for the biological parameters) women did not have any pathological process, which may have interfered with the results.  As you suggest, this may be confusing for the reader and we have deleted the “high-risk unit” from the text.

Line 130: why is "(undergraduate/university degree)" the only option for education level?

Response: Our hospital assists a population of middle/high socio-economic income. We have rarely found women with lower educational level (middle school) or illiterate. In this study, all the women had higher education (bachelor's degree or higher).

Were smoking and alcohol considered separately or was saying yes to either one considered a "yes" for the entire variable?

Response: They were independent variables. We have modified the text to make it more coherent.

I am skeptical of the use of stepwise regressions: see e.g. Smith (2018) https://journalofbigdata.springeropen.com/articles/10.1186/s40537-018-0143-6 I don't think the paper can be accepted unless this is changed. Select your covariates based on previous literature/mechanistic basis, etc.

Response: very interesting article. According to the article, the biggest problem with stepwise regression could be with the handling of Big Data. The paper also describes that stepwise regression is successful if it contains a balanced number of predictor variables and covariates. In our case, the covariates that fit our models were specially selected according to our univariate analysis (the models were build knowing, at least, the relationship with the outcome). We considered that the variables were balanced throughout the steps. In addition, although the reviewer's comment is very accurate, we wanted to check the contribution of each covariate by gestational trimester. We have clarified this aspect in the discussion (lines 541-544) and added a reference 56.

Was there any correction for multiple tests done, especially with a relatively small sample size? e.g. false discovery rate? If not, why?

Response: You are right about the comment. We did not use multiple test correction, since they are usually applied to test the differences between particular pairs of experimental groups (when K>2; “The purpose of the multiple comparison methods is to control the ‘overall significance level’ of the set of inferences performed as a post-test after ANOVA or as a pairwise comparison performed in various assays”; PMID: 30157585). In our case, the number of groups (K) was 2.

Overall, I still think the paper needs some revisions. The content is very interesting and can inform future perinatal/maternal health research. I applaud the authors for their thorough consideration of biopsychosocial factors.

Response: Thank you very much again for your interesting comments, we hope we have satisfied your considerations.

Round 2

Reviewer 1 Report

I appreciate the improvements and revisions. I still have a few suggestions/comments:

  • Figure 3 was not fully shown in the first draft, so this time the data makes more sense. However, it did take a bit to interpret. I am not seeing p-values? Did you use all of the variables in the regression or just ones with a univariate significance of >.2 or something?
  • The ad-hoc scales are still a concern given these had major results/conclusions for the study. I strongly recommend validating the tool prior to using in more studies. But the reliability was adequate, which is positive.
  • It is recommended to discuss ART more throughout the intro and discussion as this seems to be the primary interest of the authors, correct? In this area of research, bio-psycho-social stress often refers to chronic stress related to racial disparities and poverty. Although a different type of stress, ART and maternal age are a type of stress and need to be woven throughout more prominently to avoid confusion. It may be important to include this in the purpose and aims. Do you measure ART? If not, you may need to only discuss the hypothesis of ART in discussion as twin delivery and age are only assumptions for ART. It seems author's hypothesis is more about the stress of working mothers and maternal age, so make the introduction stronger. It is still unclear what the main purpose, the main point, of the study is supposed to be.

Author Response

Response: Thank you for taking the time and continue improving our manuscript. Please, see below our response.

  • Figure 3 was not fully shown in the first draft, so this time the data makes more sense. However, it did take a bit to interpret. I am not seeing p-values? Did you use all of the variables in the regression or just ones with a univariate significance of >.2 or something?

Response: We are sorry for the confusion in the previous draft. Indeed, p-values are not explicitly shown in the figure. However, the graphs show adjusted OR and their 95% CI, and all bars that cut the dotted line (OR=1) are non-significant variables (all those that do not touch the dotted line will have a p-value<0.05). We used this type of graph since OR has the advantage to display the factors associated with the outcome as risk factors (OR>1) or protective factors (OR<1).

Regarding the second question, only those variables that could be associated with the outcome (p-value<0.1 in the univariate analysis) were introduced in the models since these variables can be considered modulators (Reference [35]: “The most parsimonious model is shown leaving in variables with a p-value of 0.10 or lower to be important potential confounders”).

  • The ad-hoc scales are still a concern given these had major results/conclusions for the study. I strongly recommend validating the tool prior to using in more studies. But the reliability was adequate, which is positive.

Response: We agree with this comment. We have recently published a paper in pregnancy that uses these scales (reference 28).

  • It is recommended to discuss ART more throughout the intro and discussion as this seems to be the primary interest of the authors, correct? In this area of research, bio-psycho-social stress often refers to chronic stress related to racial disparities and poverty. Although a different type of stress, ART and maternal age are a type of stress and need to be woven throughout more prominently to avoid confusion. It may be important to include this in the purpose and aims. Do you measure ART? If not, you may need to only discuss the hypothesis of ART in discussion as twin delivery and age are only assumptions for ART. It seems author's hypothesis is more about the stress of working mothers and maternal age, so make the introduction stronger. It is still unclear what the main purpose, the main point, of the study is supposed to be.

Response: The main objective of this study was not focused on ART or work context alone, but rather to report on a global model that describes women's vulnerability during pregnancy in our social context. In the studied population, racial disparities and poverty (which are key stress factors) are not relevant determinants. Instead, a key point was the delay in maternity (not only as a personal choice but also in response to demanding work), as you point out, may impose additional stress on the pregnant woman. The use of ART is a consequence of this and, we have explored this fact. However, the manuscript is not centered on ART. In fact, women with ART-derived pregnancies were quantified, but we did not classify them according to the ART technique used, since it was not the main purpose of the study. As the reviewer notes, this could be important to remark. We have included additional discussion on this aspect in lines 439-443.

Regarding the influence of maternal work stress, we did not aim to focus only on this aspect. The maternal psychological environment may be governed by many other stressors in addition to work-family conflicts, such as low availability of resources (material, economic, human), excessive concerns during pregnancy, or low resilience. However, we agree with the reviewer and we have modified the abstract, introduction, and hypothesis focused on advanced maternity age, which (the main social determinant in our context), which brings together psychosocial and biological spheres. We have also discussed maternity age as a key social component of pregnancy (lines for “social factors” paragraph).

Round 3

Reviewer 1 Report

 .